# Applying Web Augmented Reality to Unexplosive Ordnance Risk Education

**Harith A. Hussein** [1,*], **Qabas A. Hameed** [1], **Reem D. Ismael** [2], **Mustafa Zuhaer Nayef Al-Dabagh** [3] and **Moudher Khalid Abdalhammed** [4]

1  Department of Computer Science, College of Computer Science and Mathematics, Tikrit University, Tikrit 34001, Iraq
2  Computer and Informatics Centre, Tikrit University, Tikrit 34001, Iraq
3  Department of Computer Science, College of Science, Knowledge University, Kirkuk Road, Erbil 44001, Iraq
4  General Management Department, College of Administration and Economics, Tikrit University, Tikrit 34001, Iraq
*  Correspondence: harith_abd1981@tu.edu.iq

**Abstract:** Unexploded Ordnances (UXOs) are considered a global concern and a persistent hazard due to their capability to endanger civilians and the place where they are located, and the probability of remaining active explosives even after decades of ending a conflict. Hence, risk education is crucial for providing individuals with life-saving knowledge on recognizing, avoiding, and reporting UXO threats. The main objective of this study is to develop a web augmented reality (AR) application to investigate the effect of WAR on non-explosive ordnance risk education. Firstly, UXO 3D models are edited and constructed using the Blender 3D computer graphics software. Secondly, the proposed web AR application is developed using MindAR JavaScript-based library. Finally, the web application QR code and UXO Hiro codes are printed on infographics and brochures to be distributed to secondary school students aged 12 to 18 at six public and private schools in Tikrit City, Salah al-Din governorate, Iraq. Survey questions are validated and distributed to be collected from 137 respondents. The present study shows that the proposed web AR application increased respondents' information in identifying UXO by 54.7%. Approximately 70% of respondents use the Internet for more than 3 h daily. Institutions should use new risk education methods in line with the tremendous technological growth and invest students' knowledge and time in this field. Better risk education teaching methods can save lives.

**Keywords:** unexploded ordnances; UXO; risk education; MindAR; web augmented reality

## 1. Introduction

Unexploded ordnance (UXO) is any sort of war explosive remnants that have been either dropped, fired, projected, or launched during the war and remain active. Types of UXO may include landmines, which are explosive objects expected to be activated by vehicles or people, and abandoned explosive ordnance (AXO), which are explosive objects that have never been fired or dropped during a battle but remain active in the area. UXOs are considered continual threats to civilians and the economy, especially in low-income and middle-income nations [1]. The 2022 Cluster Munition Monitor Report (the 13th annual edition) states that, since 1999, the Monitor Organization has recorded 23,082 deaths from cluster bombs, including victims of both cluster munitions attacks and unexploded ordnances. Worldwide estimates for total casualties range from 56,500 to 86,500. In 2021, two-thirds of total casualties were, notably, children, particularly boys [2].

United Nations Children's Fund (UNICEF) reported that explosive remnants of war endangered over 22,000 children in eastern Ukraine only [3], while in Yemen, UXOs are the cause of over 75% of children casualties [4]. Moreover, over 125 children were injured or

killed in Yemen in 2021 because of UXOs; 52 children were killed and 73 were wounded [5]. This demonstrates an increased incidence of child casualties when compared to 2020, as the United Nations (UN) confirmed the death of six children and the wounding of 12 children as a result of UXOs.

In Iraq, decades of wars and conflicts have caused it to be the most polluted country on the globe regarding the scope of the mined regions, and the fourth most polluted country with regard to cluster munitions contamination [6]. Legacy pollution can be found along Iraq's borders with Kuwait and Iran, notably in the southern region. Internal conflicts after 2003 and the ISIS occupation from 2014 to 2017 have resulted in further UXO pollution. Consequently, UNICEF urges the Iraqi government and international donors to endorse the scale-up of UXO education programs so that children and society members obtain UXO education in schools and communities throughout Iraq's previously conflict-affected areas [7]. Despite the great efforts dedicated to the removal of UXOs and to cleaning up the grounds, in recent years, there has been an increase in the number of casualties and victims, especially among children. Figure 1 shows deminers during UXO clearance.

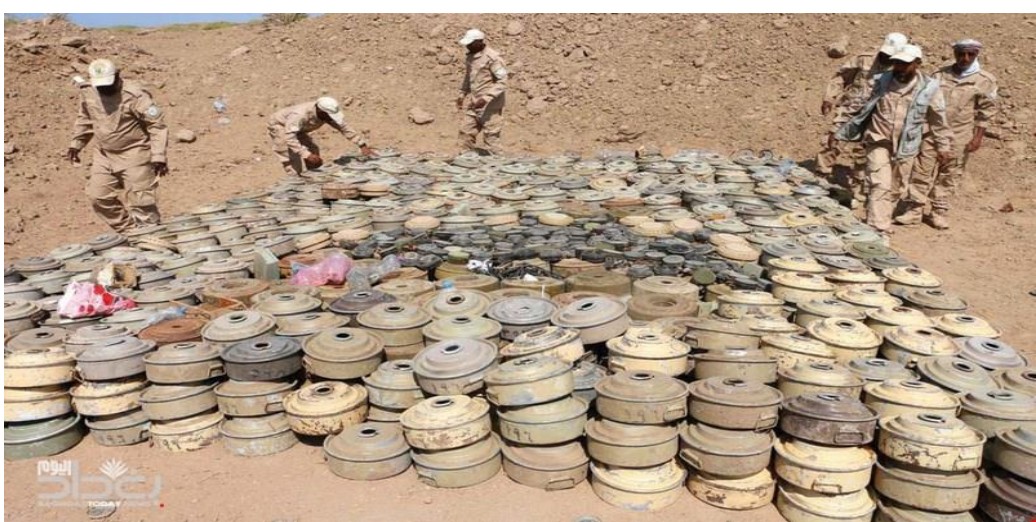

**Figure 1.** UXO deminers clean what local people call "The Valley of Death" in Iraq.

The risky effects of UXOs have resulted in efforts to develop programs related to risk education among children. Risk education is essential when a community is at risk of encountering the leftover weapons of war, and governments and organizations should provide lifesaving information. The goal of risk education is to provide as many individuals as possible with potentially lifesaving knowledge on how to identify, prevent, and report dangers. This is highly critical for children and returning societies [8]. In fact, a large number of UXO victims are children of both genders with distinctive capabilities (e.g., essential skills, different viewpoints, teaching skills) that can contribute to decrease the risks and impacts of danger. Consequently, education is required to raise awareness and encourage safe behavior among the children and UXO-affected societies. This strengthens societies' relationships with minefield operators and simplifies data gathering in the stage of the primary assessment survey [9].

Traditionally, such awareness-raising practices take the form of booklets, presentations, posters, layouts, and radio and TV broadcasts. Usually, these educational approaches are designed for schoolchildren who are supposed to remember and filter their acquired knowledge with their parents through open conversation. Nevertheless, evolving existing education and communication technologies are becoming increasingly important to engage a better new generation of learners. It is asserted that, as technology advances, individuals in the present time learn and comprehend in new ways that their predecessors do not. Furthermore, the advancement of digital technology, including artificial intelligence, computer vision, networks, and smart handheld devices, are increasingly omnipresent [10].

Extended reality (XR) technology, such as augmented reality (AR) and virtual reality (VR), has been applied and proved practical in safety education and risk management. Though the effect of AR on UXO risk education is still unknown, it is found to be effective and efficient for learning and memory encoding. Specifically, neuronal studies record approximately three times the activity level of human brains using AR in subjects exposed to specific tests [11]. Furthermore, these technologies create new possibilities to explore alternative methods of risk education by employing visualization and interaction for children and other society citizens, which allows for better engagement. While UXO danger is recognized worldwide as a challenge, efforts to develop and create effective risk education programs related to UXO seem virtually nonexistent. Accordingly, there is a genuine need to develop a risk education approach for students to receive safety knowledge conveniently.

This paper presents a UXO risk education method based on a web AR application to educate secondary school students (aged 12–18) about the danger of UXO and raise awareness of the necessity and significance of caution. It contributes to the existing knowledge on UXO risk education through proposing a web AR application based on the recently launched JavaScript-based API (mind-ar-js) for UXO risk education that makes the learning process affordable, enjoyable, and effective. It utilizes the appropriate statistical tools in order to analyze the collected data and identify the effects of web AR. This study is structured as follows: Section 2 sheds light on the current research with risk factors and presents web AR solutions. Our web AR application and the design of the experiment are shown in Section 3. In Section 4, the analysis and the results of the experiment are discussed.

## 2. Research Background

### 2.1. Technology in Risk Factor

Technology brings processes, tools, and information exchange to promote development for the betterment of humanity. It helps people to solve their problems, reduce excess time, save lives, and to interact and exchange information. A series of recent studies suggest technology-based solutions that can minimize or prevent the risk of natural and human-caused hazards. Moreover, previous studies have shown that gaming and augmented reality are considered practical tools in risk education.

As for gaming, the study of [12] emphasizes the importance of the ministry of education in undertaking prompt and guiding roles in support for DRR efforts through the curricula in schools. This is achieved by directing thinking and discourse toward the complex reasons that cause disasters. Furthermore, the study recommends that video games can be effective teaching instruments for instructors, and when used in combination with other methods to promote participation, they may offer a way to increase students' understanding of disaster preparedness and response.

Concerning natural disaster risk management, many studies have presented different approaches to inform users of different ages about how to behave with risk situations. The study of [9] proposed an educational game based on the Android platform to mitigate earthquake risks augmented with video of earthquake stages. This proposed method is divided into two parts: an educational game and an earthquake simulation. Technological advancements, particularly mobile devices, have altered the landscape of education. Entertaining and joyful games will make learning operations easier and meet most of a child's basic needs. In [13], the authors implemented an innovative adaptive e-learning gaming environment, which is capable of combining personalization, communication, and emotions called "ALICE", to eliminate the traditional limitations in earthquake risk education programs. In addition to combining a traditional and gamified approach, AR/VR improved the popularity of gamified applications [14]. The study of [15] focused on augmented reality (AR) and head-mounted displays (HMDs) to implement an AR based system designed for children who find it difficult to understand disaster education. Hence, instead of teaching disaster education to children, adults should properly instruct them to take immediate

actions in the event of a disaster. This type of learning is referred to as Immediate Action Commanders (IACers), and is important for technology-enhanced IACer training programs with high situational and audio-visual realities. In addition, the systems are designed to realize voice-based interactions between the virtual objects and the trainee.

As for human-caused risks, such as mine risk, an MRE (mine risk education) app presents a platform to provide children with a standardized quality MRE that they can access frequently. By playing this game, children can practice behaviors and thought processes in a simulated environment that have real-life applications. In addition to dynamically engaging young learners, it also facilitates learning for adult educators as well. The five stages of the app are corresponding to lessons. These lessons cover important MRE topics. Another key feature of this app is a back-end system to track basic demographic details about users as well as their performance and progress through the five stages [16].

Another common risk site is the construction site, because of its complexity, where accident rates are still high. Hence, the study of [17] introduced a mobile-based construction safety education framework; the system is based on AR-VR technologies that aim to enhance conventional construction safety education, thus resolving the limitation in the existing pedagogical approaches and means at the tertiary class, which are incapable of providing students with practical and realistic safety knowledge. The proposed system's objective is to encourage construction students to extend their safety learning and risk identification abilities. The outcomes of the system concluded that employing mobile-based AR-VR solutions would enhance construction safety effectively.

As for road accident risk, a VR system dedicated to road safety education and training for children was presented. This system uses the concepts of a VR game set to allow the children to learn about traffic laws and practice them at home without any threat of exposure to the real-world environment. In addition, the suggested system evaluates the general students' performance in the virtual domain to develop the skills of the children regarding their road-awareness abilities. Finally, the experimental outcomes demonstrate the system's positive impact in enhancing children's road-crossing behavior [18]. Another safety road education and prevention application for children based on VR was proposed. The system consists of a detailed scenario specially designed for children risk prevention in urban places. Then, the scenario was experimented in a VR environment. The proposed system experiment demonstrated positive results regarding system effectiveness interactions from functional and cognitive viewpoints [19].

Another study [20] used a different digital technology, the cloud-based education platform. This study described a practical research project conducted in Muroran City, Japan, which aimed to involve high school students in a disaster education program through digital technology. People in disaster-prone areas will be able to receive tailored warnings and behavioral guidance once such digital personalized services are available. The researcher believes this can lead to a more encompassing, secure, adaptable, and sustainable community and city in which people's ability to deal with the unexpected evolves over generations. However, to our knowledge, no prior studies have examined the effect of using augmented reality in UXO risk education. Schools can play an essential role in improving educational schemes for disaster reduction and developing didactic methods. Furthermore, educational institutions need to take advantage of the tremendous technological growth and invest in students' knowledge in this field.

### 2.2. Web AR Solutions

Technological progress and rapid development of various technologies allow for better tools and methods to build efficient web AR. These advancements describe a significant possibility to perform and implement adequate AR services and functionality. The main approaches to building web AR applications can be categorized into two primary fields: JavaScript API and browser-based AR. The following section introduces a number of examples of each category.

JavaScript APIs: This implementation approach fulfils the cross-platform condition of building web AR applications, which are independent of the specifications of the device.

- AR.js is one of the available options for web AR development that requires minimal effort. Web AR frameworks based on JSARToolKit5 and Three.js can be performed on various platforms and different browsers with WebGL and WebRTC. Currently, AR.js can only work with the fiducial marker since it is capable of performing uncomplicated matrix operations [21].
- MindAR is another open-source JavaScript dedicated to web AR. The library can support both face tracking and image tracking. In addition, integration with the Three.js library is supported by MindAR. Finally, a web compilation tool is also provided as an Image Targets Compiler [22].
- Argon.js is a JavaScript framework for integrating AR content into web applications. It was initially designed to use the AR features of iOS's Argon AR web browser, facilitating the process of providing mobile AR experiences, though now it supports different mobile platforms [23].
- Awe.js is a proprietary library used on the awe.media platform to develop web-based immersive reality (mixed reality, augmented reality, interactive 360° scenes, virtual reality), as well as awe image web AR, face filters web AR, spatial web AR, and location-based web AR [24].

Browser based AR: A web browser with built-in AR functionality has a lot potential areas of use and is easy to use. The browser-based AR's primary advantage is simplicity and speed, resulting in a more significant usage.

- Argon: One of the first attempts at this approach was argon. It was created as a platform for deploying widespread AR applications that use a pre-defined URL to access AR services. Argon web browser includes many features, such as the capacity to render augmented reality media designed with JavaScript framework argon.js [23].
- WebXR API device: WebXR is a set of standards employed jointly to support the generation of 3D scenes to hardware created to deliver virtual reality or to merge graphics and different digital elements to the real world (AR). This WebXR implements the foundations of the WebXR feature group, supervising the choosing of output devices, generating the 3D settings to the selected device at the proper frame rate, and controlling motion vectors through utilizing input controls [25].
- BlippAR browser: A browser-based AR released in 2011. With the browser functionality, the user can experience the augmented reality using image recognition, image tracking, and object recognition. The BlippAR browser supports both Android and IOS [26].
- Wikitude browser: A browser-based AR released in 2008. With the browser functionality, the user can experience the augmented reality using location-based AR, geolocation technology, image recognition and tracking. The Wikitude browser supports both Android and IOS [27].
- WebARonARCore: An experimental browser targeting Android OS. The browser allows developers to develop AR experiences with the help of web technologies. WebARonARCore is built based on two essential technologies, Chromium and Android ARCore SDK. However, it is not a full-featured web AR browser. WebARonARCore is only designed to allow developer investigation [28].

## 3. Materials and Methods

### 3.1. Construction of Explosive Ordnance 3D Models

A model is a "three-dimensional representation of an existent person or thing or of a suggested structure" [29]. For more than a thousand years, models have been used to instruct people. With the advent of three-dimensional (3D) computer-generated graphics that are visually appealing and frequently interactive, technology has begun to challenge this conventional method of instruction lately. It has taken a lot of work to establish 3D

construction processes, which may be carried out using either conventional surveying or cutting-edge 3D modeling software. In the 2000s, the first research on augmented reality using the 3D product model emerged [30]. The 3D models used in this study are created, edited, and modified using Blender software. Blender is a free and open-source 3D computer graphics software. It provides a strong toolset that is capable of creating 3D models, animated 3D characters, animated 3D films, an environment for video games, and much more [31]. Figure 2 shows the editing of UXO color and dimensions using Blender.

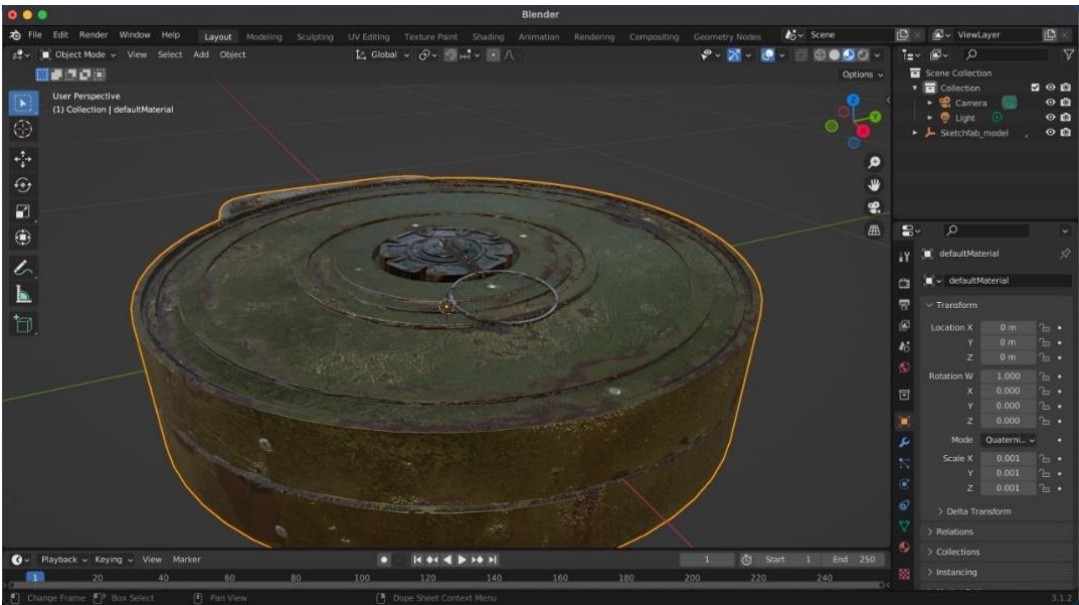

**Figure 2.** UXO 3D modeling using Blender.

To find out how well the UXO 3D models match the actual models, an interview was conducted with three experts from the UXO Treatment Division, Civil Defense Directorate in Salah al-Din governorate, Iraq. During the interview, the 3D models were displayed one after another on a data show for evaluation. The experts were asked one question: "On a scale of 1–10, does the computer-based 3D UXO model match the actual UXO?" (1 strongly disagree and 10 strongly agree). These results were recorded to calculate the mean and standard deviation (SD). The overall evaluation results are shown in the Table 1 below.

**Table 1.** The overall 3D models evaluation results.

| No. | 1 | 2 | 3 | 4 | 5 | 6 | 7 | 8 | 9 | 10 |
|---|---|---|---|---|---|---|---|---|---|---|
| UXO | Grenade | Mortar Shell | Landmine vs 50 | pmn2 | Tmi | AT Mine | Ant. Personal Mine | Tm 46 | Tm 57 | PROM-1 |
| EXP.1 | 6 | 7 | 8 | 5 | 6 | 7 | 9 | 7 | 6 | 7 |
| EXP.2 | 7 | 8 | 6 | 6 | 5 | 7 | 8 | 6 | 6 | 5 |
| EXP.3 | 5 | 7 | 7 | 8 | 6 | 7 | 7 | 5 | 5 | 8 |
| Mean | 6 | 7.3 | 7 | 6.3 | 5.7 | 7 | 8 | 6 | 5.7 | 6.7 |
| SD | 1 | 0.58 | 1 | 1.5 | 0.8 | 0 | 1 | 1 | 0.6 | 1.5 |

The table illustrates the experts' evaluation results on how well the UXO 3D models match the actual UXO models. Generally, it is obvious that the overall results are acceptable. The highest acceptance means are 8, 7.3, 7, and 7 for models 7, 2, 3, and 6, respectively. In contrast, lower acceptance means are 5.7 for both 5 and 9.

### 3.2. The Proposed Web AR Application Overview

In this study, a web AR application is proposed to educate secondary school students (aged 12–18) about the risk of UXOs and their different shapes and types. Furthermore, the proposed application can raise awareness of the necessity and significance of caution in regard to any war remains. Given that the target users of the web AR application are secondary school students under 18 years old, the usage and layout of the web AR application were designed to be simple. They require no prior training to use it. The development of the proposed web AR application is based on two main components. MindAR is a JavaScript-based API released in 2022 and three.js another JavaScript library used to display 3D models. It can run on any desktop and mobile browser with WebGL and WebRTC. The following Figure 3 shows the proposed web AR overview.

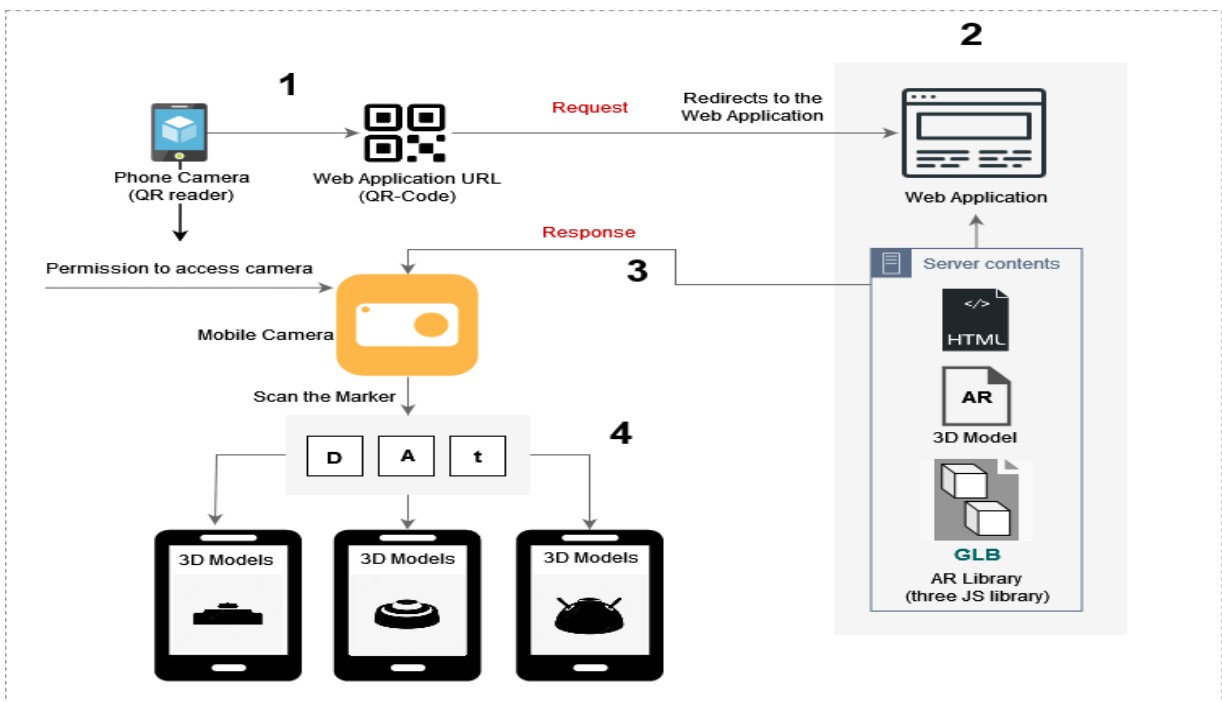

**Figure 3.** Web AR application main components overview.

To facilitate the students' access to the web AR application, the infographics and brochures (Section 3.3.3) containing a quick response (QR) code and ten Hiro markers were distributed to the schools where our experiment was implemented. First, students must point their mobile phone camera to the web AR application QR code to browse and access the proposed application. QR codes are considered a practical, engaging, and informative tool for students' informal learning [32]. Second, students are asked for their permission to access the device camera. Finally, with ease, the students can scan the Hiro markers. After successfully detecting the Hiro marker, the UXO 3D model is rendered and displayed on the mobile screen. Students can view and rotate the UXO models where each Hiro marker is associated with different UXO 3D models.

The Hiro markers have predefined form, color, dimensions, and properties employed as anchors where the AR experience occurs [33]. Furthermore, they can achieve excellent robustness and accuracy in varying environmental requirements.

### 3.3. Experimental Design

An experimental design means testing objectives by a set of procedures [34]. It is the blueprint of the procedures enabling the researcher to test the stated objectives and reaching valid conclusions about the relationships between independent and dependent variables [35]. The implemented experimental design includes the following points:

1.  Determining the study's target age group, population size, and sample.
2.  Designing questionnaire components and survey questions.
3.  Testing the questionnaire form by a group of experts.
4.  Distributing infographics and brochures to the visited schools.
5.  Distributing questionnaire form.
6.  Utilizing the appropriate statistical tools in order to analyze the collected data and obtain the final results.

### 3.3.1. Population and Sampling

The current study's population includes 11,451 male and female students, representing all the secondary schools' students who are studying at seventeen public and private schools in Tikrit City, Salah al-Din governorate, Iraq. Visited schools were coordinated with the Civil Defense Directorate in Salah al-Din, UXO Treatment Division, from 17 February 2022 to 22 April 2022 during the second semester of the academic year 2021–2022. The following Table 2 shows the total population and this study sample.

**Table 2.** The population and sample of the study.

| No. of Secondary Schools | No. of Population | | No. of Visited Secondary Schools | No. of Sample | |
|:---:|:---:|:---:|:---:|:---:|:---:|
| | Male | Female | | Male | Female |
| 17 | 6757 | 4694 | 6 | 1986 | 1320 |
| | Total | | | Total | |
| | 11,451 | | | 3306 | |

### 3.3.2. Questionnaire Design and Test

In the last few years, social researchers acknowledge the importance of conducting survey research with children directly, rather than relying on the findings of qualitative research and prevalence data collected by proxy from adults [36].

The survey questions of this study are tested and validated according to research methods' rules taking into consideration the respondents' ages. The survey questions include the following components.

- Demographic information: Includes relevant basic personal information that is required to identify respondents. The relevant information includes gender, age (divided into two sections, 12–14/15–17), and daily Internet use.
- Section One: Knowing UXO risks: Includes the survey questions developed to measure the independent variable knowing UXO risks. Four statements are given to measure the dimensions of this variable. This section aims to know the extent to which the respondents comprehend the risk of unexploded materials and their understanding of the consequences of the explosion. This section also intends to find out the influence of the respondents on their colleagues and family members.
- Section Two: Presence in risky places: Includes the survey questions developed to measure the presence in risky places variable. Three statements are given to measure and cover the dimensions of this variable. This section aims to find out the presence of the respondents in places contaminated with unexploded materials or war remnants, with or without their knowledge.
- Section Three: Identify UXO: Includes survey questions developed to measure respondents' ability to identify UXO and war remnants. Three statements are given to measure the dimensions of this variable. This section aims to find out the respondents' current level of UXO risk knowledge and how easily these UXOs are identified by them.
- Section Four: Current UXO risk education source: Includes survey questions developed to measure respondents' current UXO risk education source. This section aims

to know the respondent's current UXO risk education source, if any. Three statements are given to measure the dimensions of this variable.

- Section Five: User experience: This section aims to find out the extent of the user's knowledge of the AR concept and how fast web AR application responds. This variable also aims to find out the significances of the three-dimensional models and to indicate the extent to which the experiment contributes to increasing the respondents' knowledge of the possibility of identifying unexploded materials. It includes survey questions developed to measure respondents' experience. Five statements are given to measure the dimensions of this variable.

The survey questions were tested before distributing the questionnaires to the respondents in order to ensure its validity and ability to measure the variables of this study, to ensure that the questions are understood and answers are accurate. The questionnaire form was presented to a group of statistics experts, AR experts, and specialists in the field of UXO risk education, and their names and addresses are indicated in File S1. This is done in order to find out experts' opinions on the questionnaire content, its ability to measure the study variables, to determine the clarity and accuracy of statements linguistically and scientifically, and to be understandable to the respondents. Several revisions and corrections were made to the questionnaire before it became valid for distribution according to the opinion of the experts. Thus, the final form is taken to be as in File S2. Figure 4 shows the survey question components.

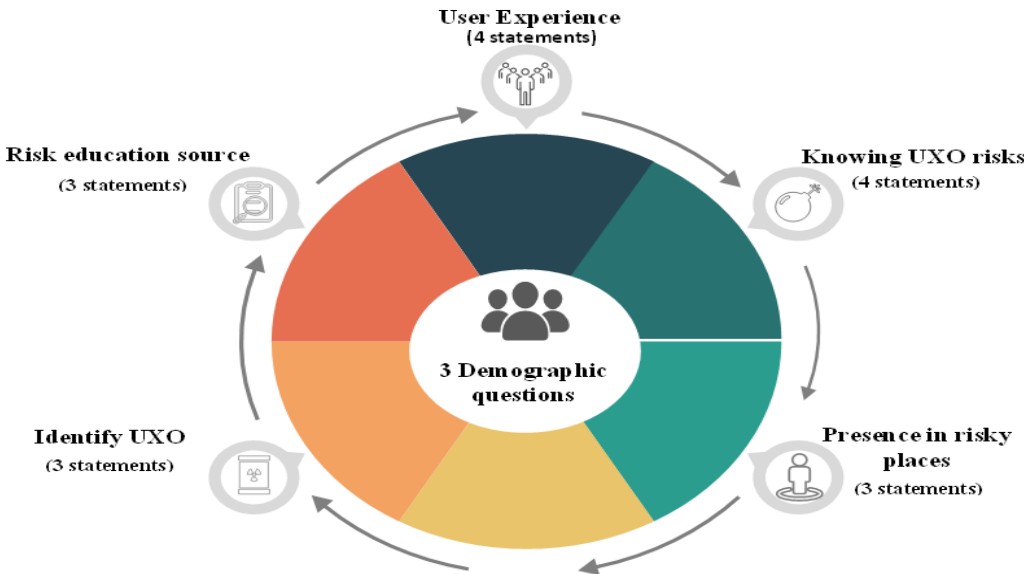

**Figure 4.** Overview of questionnaires components.

This study adopted the Likert triple scale, which grades according to the measuring statements (agree, not sure, and disagree) with the numerical scale 1, 2, 3, respectively. The researcher considers children's limited understanding of Likert response formats [37].

### 3.3.3. Distribution of Infographics and Brochures

The infographics and brochures were distributed in association with the Civil Defense Directorate in Salah al-Din/UXO Treatment Division. Researchers conducted more than 30 visits to secondary and high schools in Tikrit, Salah al-Din, Iraq from 17 February 2022 until 22 April 2022. The Civil Defense Directorate in Salah al-Din uses risk education infographics and brochures provided by the international and local UXO organizations operating in Iraq, for example the Iraqi Mine/UXO Clearance Organization (IMCO), MAG, and the International Committee of the Red Cross (ICRC).

This study used risk education documents provided by ICRC (Files S3 and S4). The documents are divided into two types: infographics and brochures. Infographics were

installed in public areas inside the schools where this study was conducted. A bring your own device (BYOD) policy is allowed in these areas. This is an excellent opportunity for students to experience the web AR application, especially as no pre-installation is required and there are no use restrictions for the platform/device. Brochures were distributed directly to the students, allowing them to experience the web AR risk education application with their family and friends. The following Figure 5a,b represent students' browsing of both types.

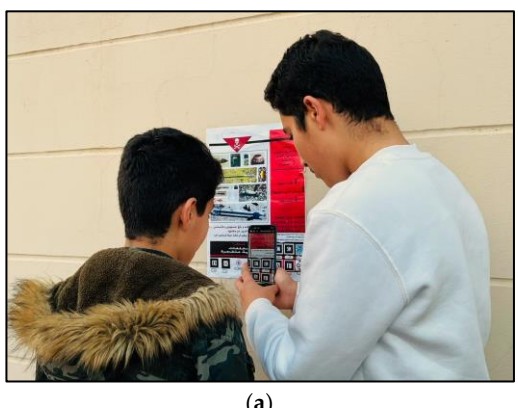 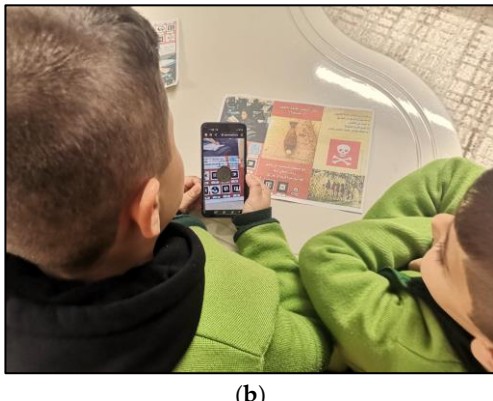

(**a**)                (**b**)

**Figure 5.** (**a**) Browsing UXO on infographic. (**b**) Browsing UXO on brochures.

The survey questions were distributed in two ways. Paper-based forms were submitted to students directly during the risk education session. An electronic version was created using Google Forms. Students were asked to scan the QR code and contribute to the study. The following Table 3 shows the final valid questionnaire forms according to both ways.

**Table 3.** The final valid questionnaire forms according to both ways.

| Total Retrieved Forms | Paper-Based Forms | E-Based Forms | Total Valid Form |
|:---:|:---:|:---:|:---:|
| | 87 | 104 | |
| 191 | Valid forms | Valid form | 137 |
| | 48 | 89 | |

The valid forms were utilized using the appropriate statistical tools in order to analyze the collected data and obtain the results, which will be presented in the following section.

### 3.4. Statistical Description

The researchers used SPSS (Statistical Package for the Social Sciences) to analyze the data of this study. After being coded, the respondents' data were inserted to the above program. Then, the results of each of the five factors were calculated according to gender and age groups. Mean, Std. deviation, frequency, and percentage were calculated according to the questions of each of the five factors. Similarly, the process was repeated with the factor "using the Internet". The resulting tables are displayed in the following sections, and the rest are shown in the Supplementary Materials at the end of this study.

### 4. Results and Discussion

This section describes the variables through which the study sample participants are distributed. Table 4 shows the distribution of the study sample participants by gender, which varies in proportion between males and females. The percentage of males in the surveyed sample was 59.1% compared to that of females, which amounted to 40.9%. As

for the distribution according to age variable, the highest age group was 15–18 years old, representing by 59.9%, while the lowest age group was 12–14 years old, representing 40.1%.

**Table 4.** Respondents' gender and age groups.

| No. of Respondents | | Gender | Age Groups | | Total |
|---|---|---|---|---|---|
| **Male** | **Female** | | **12–14** | **15–18** | |
| 81 (59.1%) | 56 (40.9%) | Male | 33 | 48 | 81 |
| | Total | Female | 22 | 34 | 56 |
| | 137 | Total | 55 (40.1) | 82 (59.9%) | 137 |

Table 5 shows the distribution of the study sample by age group and Internet use. It was found that the highest percentage was of those who used the Internet more than three hours a day, 69.3%, followed by 23.3% for those who used it between one and three hours. In contrast, the lowest percentages were of those who did not use the Internet daily or used it for less than an hour, 4.4% and 3%, respectively.

**Table 5.** Respondents' Internet use and age groups.

| Not Daily | | <1 h | | 1–3 h | | >3 h | |
|---|---|---|---|---|---|---|---|
| **12–14** | **15–18** | **12–14** | **15–18** | **12–14** | **15–18** | **12–14** | **15–18** |
| 2 | 4 | 1 | 3 | 18 | 14 | 34 | 61 |
| Total | | Total | | Total | | Total | |
| 6 (4.4%) | | 4 (3%) | | 32 (23.3%) | | 95 (69.3%) | |
| Total | | | | | | | |
| 137 | | | | | | | |

Table 6 summarizes the results of the first factor (knowing UXO danger), where the statements X1–X4 indicate agreement and good levels for the statements of this factor. The percentages according to the overall indicator indicate that 73.7% of the sample participants agreed on the content of the statements of this factor. In comparison, the percentage of those who were not sure was 21%, while that of those who did not agree was 5.3%. These percentages had an arithmetic mean of 1.35. The initial significance of the results of the first factor is that there is a general understanding among school students of the dangers of unexploded remnants of war and their impact on family members and friends.

**Table 6.** Summary of respondents' answers on knowing UXO risk.

| Sample | Likert Scale | | | | | | Median |
|---|---|---|---|---|---|---|---|
| | Agree | | Not Sure | | Don't Agree | | |
| | **Total** | **%** | **Total** | **%** | **Total** | **%** | |
| X1 | 103 | 75.2 | 32 | 23.4 | 2 | 1.5 | 1.26 |
| X2 | 117 | 85.4 | 17 | 12.4 | 3 | 2.2 | 1.17 |
| X3 | 61 | 44.5 | 55 | 40.1 | 21 | 15.3 | 1.7 |
| X4 | 123 | 89.9 | 11 | 8 | 3 | 2.2 | 1.3 |
| Total | 73.7% | | 21% | | 5.3% | | 1.35 |

Table 7 presents the results of the second factor (presence in risky places) for all statements (X5–X7) with comparable levels. The percentages according to the overall indicator indicate that 33.1% of the respondents agreed on the content of the statements of

this factor. In comparison, the percentage of those who were not sure was 26.5%, while the percentage of disagreement reached 40.4%. These results came with an arithmetic mean of 2.07. The results indicate that nearly half of the sample members were previously present in places with dangers and remnants of war.

**Table 7.** Summary of respondents' answers on "presence in risky places".

| Sample | Likert Scale | | | | | | Media |
| | Agree | | Not Sure | | Don't Agree | | |
| | Total | % | Total | % | Total | % | |
|--------|-------|------|-------|------|-------|------|-------|
| X5 | 70 | 51.1 | 37 | 27 | 30 | 21.9 | 1.71 |
| X6 | 36 | 26.3 | 41 | 29.9 | 60 | 43.8 | 2.17 |
| X7 | 30 | 21.9 | 31 | 22.6 | 76 | 55.5 | 2.35 |
| Total | 33.1% | | 26.5% | | 40.4% | | 2.07 |

The results of statements X8–X10, clarified in Table 8, show agreement and good levels regarding this factor. The percentages according to the overall index indicate that 56% of respondents agreed on the content of these statements. While that of those who were not sure was 30.4% and the percentage of disagreement was 13.6%. These results came with an arithmetic mean of 1.57. At the micro level of each indicator measuring this factor, the index X10 reached the lowest acceptance rate, indicating that more than half of the sample cannot easily distinguish remnants of war. The highest percentage was for the index X8, which was 67.2%. This indicates that a high percentage of the sample had previously seen explosive remnants of war.

**Table 8.** Summary of respondents' answers on "identify UXO".

| Sample | Likert Scale | | | | | | Median |
| | Agree | | Not Sure | | Don't Agree | | |
| | Total | % | Total | % | Total | % | |
|--------|-------|------|-------|------|-------|------|--------|
| X8 | 92 | 67.2 | 20 | 14.6 | 25 | 18.2 | 1.52 |
| X9 | 73 | 53.3 | 48 | 35 | 16 | 11.7 | 1.56 |
| X10 | 65 | 47.4 | 57 | 41.6 | 15 | 10.9 | 1.64 |
| Total | 56% | | 30.4% | | 13.6% | | 1.57 |

The results of statements X11–X13, shown in Table 9 and indicating the answers of the sample and the overall levels, that secondary school students in the city of Tikrit agreed to a large extent with the items of this factor. These answers demonstrate that a large percentage of the sample participants had previously seen advertisements about learning risks through infographic publications, followed by advertisements on social networking sites, and, lastly, through local radio. The highest acceptance rate was found for paragraph X11 (78.1%), which indicates that a large percentage of school students had already seen paper advertisements about learning hazards that would reduce the risk of exposure to explosive remnants of war. This is followed by the percentage of participants that had viewed advertisements on social media sites and electronic sources, which reached 72.3%, and the lowest rate was about learning risks through local radio.

**Table 9.** Summary of respondents' answers on "UXO risk education source".

| Sample | Likert Scale | | | | | | Median |
|---|---|---|---|---|---|---|---|
| | Agree | | Not Sure | | Don't Agree | | |
| | Total | % | Total | % | Total | % | |
| X11 | 107 | 78.1 | 20 | 14.6 | 10 | 7.3 | 1.62 |
| X12 | 99 | 72.3 | 28 | 20.4 | 10 | 7.3 | 1.83 |
| X13 | 87 | 63.5 | 31 | 22.6 | 19 | 13.9 | 1.76 |
| Total | 71.3% | | 19.2% | | 9.5% | | 1.73 |

The results of statements X14–X18, presented in Table 10, show that the responses of the sample participants at the macro level indicate a varying acceptance rate between strong and weak. The overall acceptance rate for the statements of this factor was 44.4%, followed by 46% for those who were not sure, while the percentage of disagreement was 9.6%. The results indicate that the results of this factor are acceptable, as its arithmetic median was 1.47. At the micro level for each of the indicators measuring this factor, the highest acceptance rate was for the statement X18, which reached 54.7%. This rate reflects that more than half of the sample agreed that the proposed application contributed, to an acceptable percentage, to increasing the information of school students in distinguishing unexploded remnants of war. In addition, 52.6% of respondents agreed on their prior knowledge of augmented reality technology, which is a good percentage, perhaps due to the availability of augmented reality in most smartphone cameras and social media applications. Furthermore, 35% of respondents agreed on the ease of use of the web application and the speed of its response. Regarding the clarity of 3D models, 44.5% of the respondents agreed with this statement, and the percentage of those who were unsure was 49.6%.

**Table 10.** Summary of respondents' answers on "user experience".

| Sample | Likert Scale | | | | | | Median |
|---|---|---|---|---|---|---|---|
| | Agree | | Not Sure | | Don't Agree | | |
| | Total | % | Total | % | Total | % | |
| X14 | 72 | 52.6 | 51 | 37.2 | 14 | 10.2 | 1.64 |
| X15 | 48 | 35 | 68 | 49.6 | 21 | 15.3 | 1.55 |
| X16 | 48 | 35 | 79 | 57.7 | 10 | 7.3 | 1.32 |
| X17 | 61 | 44.5 | 68 | 49.6 | 8 | 5.8 | 1.36 |
| X18 | 75 | 54.7 | 49 | 35.8 | 13 | 9.5 | 1.5 |
| Total | 44.4% | | 46% | | 9.6% | | 1.47 |

From the above discussion of results, the following findings were reached:

1. Approximately 70% of primary school students in the study area use the Internet for more than 3 h a day. Therefore, the Civil Defense Directorate in Salah al-Din should use new risk education methods in line with the tremendous technological growth and invest students' knowledge and time in this field.
2. There is a good level of understanding among school students toward the risks of unexploded remnants of war and their impact on family members and friends. This result is inconsistent with [38], which found that youth with lower perceived self-efficacy believed that a UXO accident would not likely result in severe consequences.
3. Nearly, half of the respondents coexist in places contaminated with UXO and risky war remains. This result agrees with the statistics of local and international UXO clearance organizations.

4. Around 47% of the respondents find it difficult to identify the shapes of UXO. This result can enhance the importance of this study and open up for the use of technology in the field of UXO risk education.

5. Approximately 52.6% of respondents have prior knowledge of augmented reality technology. In contrast, the current UXO risk education sources are infographics and social media.

6. The proposed web AR application has increased respondents' information in identifying UXO by 54.7% of the total respondents.

7. Based on the positive results of the research, received through the respondents, it can be concluded that the application is useful and can be used by other countries since the danger of UXO pollution is a similar issue in other countries. In addition, the most reliable and widespread method of UXO risk education is educating school students. Furthermore, the web app requires only markers and an Internet connection, which improves the ease of use and eliminates the need for training before using the application, hence, allowing the web app to become applicable in different countries.

The challenges and limitations of the current study includes the difficulty of using the application, especially for the 12–14 age group, as the researchers could not reach all the study respondents to train them how to use the web AR application. The UXO 3D models were not acceptable or clear enough for 55.5% of the respondents (the sum of not sure and do not agree answers). The disparity in Internet access made the web AR application response different in different parts of the study area, and this is what the researchers found as well. In sum, institutions and schools with no resources can be afforded with the proposed web AR application, and the depiction of UXO 3D models can be used to teach students within a short period of time about different types and shapes of UXO.

Future research should consider the potential effects of developing web AR based on educational games. Gaming can positively affect UXO risk education outcomes.

**Supplementary Materials:** The following supporting information can be downloaded at: https://www.mdpi.com/article/10.3390/computers12020031/s1. File S1: List of experts who reviewed and validated the questionnaire form. File S2: Sample of questionnaire form. File S3: UXO infographics. File S4: UXO brochures.

**Author Contributions:** Supervision, conceptualization, methodology, and writing—original draft preparation, H.A.H.; web application development and writing—original draft preparation, Q.A.H.; data curation and writing—original draft preparation, R.D.I.; 3D modeling, M.Z.N.A.-D.; Statistics and results, M.K.A. All authors have read and agreed to the published version of the manuscript.

**Funding:** This research received no external funding.

**Data Availability Statement:** All data were presented in the main text.

**Acknowledgments:** We would like to express our thanks to the Civil Defense Directorate in Salah al-Din- Iraq/UXO Treatment Division for their support in providing all information required, documentation, and statistics to accomplish this research.

**Conflicts of Interest:** The authors declare no conflict of interest.

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
