# Peer review of "Applying Web Augmented Reality to Unexplosive Ordnance Risk Education"

_computers, doi:10.3390/computers12020031_

Round 1

Reviewer 1 Report

This article, presents an Augmented Reality web application for the Unexploded Ordnance risk education. I think that the experiment, lacks of the objective evaluation. For example, it could have been developed an educational game with the use of QR codes, and thus variables like a score or time would make possible the measurement of the quality of the AR app and the impact on the education process.   So, it would be better if you explain why you didn’t use edutainment methods/techniques or gamification which will help you for an objective evaluation of education process. Are there special reasons? Is it a topic that will concern you in future work? I understand that it is a long evaluation process which has already been completed.  

Reviewer 2 Report

This is, from my point of view, a very relevant application of xR technologies. I would only suggest the authors to provide a reference for the claim stated in line 55.

Reviewer 3 Report

This is a very interesting paper about educating young students detecting and be aware of unexploded ordnance. The paper is well written. There is lot of related work. The authors have conducted a survey that proves that their application is useful in their country.

In order to improve the quality of their paper, authors are proposed to:

-          explain in more detail how their idea differs from related work

-          provide more details about the development and the functionality of the application (and its innovating technical features) in text and by providing screenshots of the application and images during the execution of the application

-          discuss if/how their application could be useful / generalized / standardized in similar settings in other countries
